# Critical Review in Designing Plant-Based Anticancer Nanoparticles against Hepatocellular Carcinoma

**DOI:** 10.3390/pharmaceutics15061611

**Published:** 2023-05-29

**Authors:** Aalok Basu, Thanaphon Namporn, Pakatip Ruenraroengsak

**Affiliations:** Department of Pharmacy, Faculty of Pharmacy, Mahidol University, 447 Sri-Ayutthaya Rd., Rajathevi, Bangkok 10400, Thailand

**Keywords:** hepatocellular carcinoma, liver cancer, tumor microenvironment, phytochemicals, plant bioactives, nanomedicines, cancer drug targeting, nanoparticles, preclinical trials

## Abstract

Hepatocellular carcinoma (HCC), accounting for 85% of liver cancer cases, continues to be the third leading cause of cancer-related deaths worldwide. Although various forms of chemotherapy and immunotherapy have been investigated in clinics, patients continue to suffer from high toxicity and undesirable side effects. Medicinal plants contain novel critical bioactives that can target multimodal oncogenic pathways; however, their clinical translation is often challenged due to poor aqueous solubility, low cellular uptake, and poor bioavailability. Nanoparticle-based drug delivery presents great opportunities in HCC therapy by increasing selectivity and transferring sufficient doses of bioactives to tumor areas with minimal damage to adjacent healthy cells. In fact, many phytochemicals encapsulated in FDA-approved nanocarriers have demonstrated the ability to modulate the tumor microenvironment. In this review, information about the mechanisms of promising plant bioactives against HCC is discussed and compared. Their benefits and risks as future nanotherapeutics are underscored. Nanocarriers that have been employed to encapsulate both pure bioactives and crude extracts for application in various HCC models are examined and compared. Finally, the current limitations in nanocarrier design, challenges related to the HCC microenvironment, and future opportunities are also discussed for the clinical translation of plant-based nanomedicines from bench to bedside.

## 1. Introduction

Liver cancer is the third leading cause of cancer-related deaths worldwide and the most widely occurring type of cancer in Asia [1]. Apart from cholangiocarcinoma, hepatocellular carcinoma (HCC) is the major category of liver cancer; it accounts for 85% of liver cancer cases globally, with high morbidity and mortality. There are significant differences in the incidence of HCC across different genders, ethnicities, races, and geographical regions, with the incidence especially high across Asia and Africa [2]. HCC may be the terminal result of chronic liver conditions, starting from liver fibrosis, cirrhosis, and finally malignancy. Approximately 80% of patients diagnosed HCC have poor prognosis [3]. While cirrhosis is a major risk factor for liver cancer in humans, chronic hepatitis B and hepatitis C viral infections are also among the established underlying causes of HCC [4]. Although neonatal hepatitis B vaccination has now been recommended in most countries as part of the global strategy to alleviate HCC burden by 2030, the vaccine coverage in underdeveloped areas is considerably poor, and complete prevention of viral infection is thereby not possible [5]. Exposure to toxins, alcohol, and contaminated foods and the presence of metabolic diseases (such as non-alcoholic fatty liver) have also contributed to HCC development [6,7]. At the cellular level, the disease pathogenesis is, however, complex and involves several molecular failures, such as cell cycle deregulation, chromosomal instability, immunomodulation, epithelial-to-mesenchymal transition, microRNA dysregulation, and increases in HCC stem cell populations [8].

Liver transplantation, surgical removal, Y-90-based radiotherapy, transarterial chemoembolization, and percutaneous ablation have been mostly effective in the early stages of HCC (Barcelona Clinic Liver Cancer (BCLC), Barcelona, Spain, stage A), when the tumor or the lesion is less than 2 cm in size [9,10]. In fact, HCC can be cured with a good long-term prognosis, if detected early. Despite several surveillance protocols and recommendations, more than two-thirds of the patients are diagnosed during the advanced stages (BCLC stage C), when curative treatments often fail. Destruction of cancer cells and inhibition of their proliferation through chemotherapy are consequently the requisite needs of most patients. A large number of clinical trials in recent years has led to the approval of multiple drugs by the Food and Drug Administration (FDA, Figure 1). In 2007, the Sorafenib Hepatocellular Carcinoma Assessment Randomized Protocol (SHARP) and Asian-Pacific trials were conducted with 602 and 226 participants, respectively. Sorafenib was consequently approved as a first-line drug in inoperable HCC cases [11,12,13]. Sorafenib is a dual aryl urea multi-kinase inhibitor, and it exhibits strong antitumor and antiangiogenic activities. Between 2017 and 2019, the FDA permitted the use of other drugs, such as ramucirumab, cabozantinib, lenvatinib, and regorafenib, thus changing the scenario for the first line of treatment. Although there are several tyrosine kinase inhibitors now available, they only increase patient survival by two to three months [14]. More recently, the use of immune checkpoint blockage therapy has been very successful in several conditions, such as melanoma, non-small cell lung cancer, and colorectal cancer. This approach, however, is in its infancy, and many phase I and phase II trials are currently investigating different immune checkpoint blockers in combination with other agents or treatment strategies in HCC [15]. Based on the results from the KEYNOTE and CheckMate trials in 2017–2018, the FDA approved the use of the antibodies pembrolizumab and nivolumab as advanced-stage second-line treatments for HCC patients with sorafenib failure [16]. While pembrolizumab can be used independently for HCC treatment, nivolumab is generally used along with ipilimumab [17].

Sorafenib has an overall survival benefit of three months, while the second line of treatment, using regorafenib and carbozantinib, has managed to prolong survival to approximately 10 months [18,19]. Incidences of tumor recurrence and poor survival rates strongly persist despite the various lines of treatments and ongoing clinical trials, as well as the variations in morphological and molecular patterns in the disease render the clinical trials more challenging [20]. In fact, sorafenib is associated with mild to severe adverse reactions, including diarrhea, elevated blood pressure, and skin rashes [12]. The STORM trial, conducted on 1114 participants, showed that sorafenib failed to mitigate the recurrence after curative treatment [21]. Alternatively, new immune-modulatory therapies using immune-checkpoint inhibitors are prone to unbalance the immune system and cause adverse reactions that occasionally may be fatal [22]. Thus, there is an urgent need for the development of new therapeutic strategies based on a thorough understanding of tumor biology, mechanisms of anticancer molecules, and their delivery options.

Several molecules originating from medicinal and dietary plants have been reported to be effective against different types of cancer by prohibiting the activation of oncogenic pathways at cellular levels. Molecules such as quercetin, curcumin, resveratrol, epigallocatechin-3-gallate, and many others have been studied extensively due to their high potency, minimum toxicity, and ability to overcome drug resistance [23,24]. Specific bioactives, such as Guttiferone K (isolated from *Garcinia yunnanensis*) and safranal (isolated from *Crocus sativus*), have shown cytotoxicity against quiescent cancer cells. These cells reside in the G_0_/G_1_ phase and are usually resistant to conventional chemotherapy [25]. However, the effects have been mostly limited in vitro, especially due to the poor bioavailability and low biological half-lives of the plant bioactive compounds. The body treats these molecules as xenobiotics, and they are rapidly cleared by the reticuloendothelial system (RES). The required therapeutic levels are, therefore, difficult to achieve and result in high inter- and intrasubject variability, as well as a lack of dose proportionality. Furthermore, the compounds vary in their molecular structures, resulting in differences in their physical states, solubilities, partitioning, and chemical stability. Low aqueous solubility, poor gastrointestinal absorption, and clearance prevent pharmacological concentrations from being achieved in the target tumor and restrict the use of the majority of these phytochemicals in clinics [26]. Nanotechnology-based approaches or nanomedicines can provide avenues to circumvent plant bioactive-related limitations and help to increase bioavailability, improve cellular uptake through site-specific targeting, and accomplish steady-state concentrations of bioactives throughout the therapeutic regimen [27]. The present review initially describes the various molecular pathways of plant bioactives against HCC and highlights the recent advancements in plant-based nanoparticle formulations for HCC treatment. We further present the challenges in the design and development of plant bioactive-based nanomedicines for HCC treatment and reveal possible strategies to facilitate clinical translation.

## 2. Tumor Biology: HCC and Current Limitations of Drug Delivery Design

The liver is the largest abdominal organ, receiving its blood supply from two prominent sources: the hepatic artery and hepatic portal vein (Figure 2A,B). These blood vessels divide into finer capillaries and the liver sinusoids, ultimately leading to the lobules. The hepatocytes are a major cell population in the liver (~85%), providing primary sites for protein synthesis, metabolism, and detoxification [28]. Other important cell-types include hepatic stellate cells (HSCs), liver sinusoidal endothelial cells, and Kupffer cells, all of which contribute to the maintenance of liver homeostasis. In case of chronic injury to the liver, these cells commence a crosstalk, leading to production of fibrous collagen and extracellular matrix (ECM) remodeling factors. Consequently, liver diseases, such as HCC, typically initiate from underlying inflammation and cause significant changes in the liver’s extra- and intracellular pathophysiology, thus perturbing drug delivery. While normal liver tissues receive 80% of the blood supply from the hepatic portal vein, HCC is characterized by high perfusion from the hepatic artery. Therefore, low blood influx through the portal vein in HCC patients causes low nanoparticle penetration into the liver after systemic administration [29]. While the sinusoidal fenestrates are decreased, the nanoparticles must penetrate the endothelial barrier, the extracellular matrix (ECM), and the tumor stromal barriers to reach the HCC cells [30]. Understanding the biological barrier of the tumor is critical for the judicious selection of plant bioactive compounds and designing a new generation of nanomedicines for HCC therapy. Recent advancements in molecular biology techniques, including microarrays and high-throughput screening, have greatly improved our knowledge about the tumor characteristics and molecular mechanisms of HCC [31,32]. The tumor microenvironment consists of tumoral and non-tumoral cells, such as hepatic stellate cells, immune cells, fibroblasts, cytotoxic T-cells, and tumor-associated macrophages. These cells play significant roles in tumor progression by inhibiting antitumor responses, stimulating the development of new blood vessels (angiogenesis), and supporting the proliferation of cancer cells [33,34]. Recent emerging clinical trial data have suggested that sorafenib significantly affects neither the stages of angiogenesis nor proliferation in tumor progression; therefore, it is necessary to discover new drug candidates that can fill in this gap and hinder the disease at the molecular level [35]. There are also extracellular components, including different collagens, glycoproteins, proteoglycans, proteolytic enzymes (matrix metalloproteinases), cytokines, and exosomes, that usually maintain the tumor integrity and are sometimes involved in the development of drug resistance [36]. One of the proteoglycans, chondroitin sulfate, has shown abnormally high expression levels in HCC cells and is now widely exploited to develop targeted therapies against HCC [37,38]. Normal tissues have microvascular endothelial gap density and an integral structure that prohibits any macromolecules from traversing through the blood vessel walls. However, solid tumors are characterized by: (i) leaky blood vessels; (ii) wide vascular wall spaces; (iii) poor structural integrity; and (iv) a lack of lymphatic reflux. This phenomenon, related to the “enhanced permeation and retention” or EPR effect described by Matsumura and Maeda in 1986, became an attractive feature permitting passive tumor targeting of nanomedicines (Figure 2B and Figure 3A). General studies in cancer nanomedicines revealed that nanoparticles in the size range of 40 to 400 nm can ensure longer circulation times and higher tumor deposition [39,40]. The liver architecture, however, can include inherently leaky vessels, along with the abnormal vasculature arising from the presence of chronic liver diseases. Hence, proper selection of nanoparticle sizes, charges, and surface properties is essential for designing nanoparticles capable of passive entry into the HCC tumor.

Tumor heterogeneity is another major factor that causes dynamic reprogramming of the tumor microenvironment and variable expression of therapeutic target proteins throughout the disease progression, thus leading to the development of resistance in HCC [41]. Heterogeneity may occur between the tumor nodules of the same patient (intertumor heterogeneity) and that between the different locations of the same tumor node (intratumor heterogeneity) [42]. Although the advent of cutting-edge single-cell and multi-region sequencing technologies has made the genetic landscape characterization of HCC heterogeneity possible, higher intratumor heterogeneity always decreases the success rate of precision medicine and other targeted delivery systems [43,44].

There are several receptors and proteins present all over HCC tumors and on normal liver cells, such as hepatocytes, Kupffer cells, HSCs, and sinusoidal endothelial cells. Receptors expressed on hepatocytes, including asialoglycoprotein receptors (ASGPRs), transferrin receptors, folate receptors, and heparan sulfates, may serve as specific docking sites for nanoparticles (Figure 2C). Other receptors, including C-X-C type 4, low-density lipoprotein, and scavenger receptors, have also been observed. Asialoglycoprotein or “Ashwell-Morell” receptor was one of the earliest studied hepatic cell receptors, and it is capable of binding with glucosamine and galactose residues. Studies confirmed that ASGPRs are also distributed on the plasma membrane of HCC cells, and its specific expression has been considered for cell-specific drug-delivery approaches. Some popular ligands used for ASGPR-specific targeting are the carbohydrate-based polymers, such as arabinogalactan, pullulan, pectin, and dextran [45]. Transferrin receptor 2 (TfR2) is a transmembrane glycoprotein receptor belonging to the transferrin family, and it is mostly expressed in rapidly proliferating cancer cells [46]. Transferrin receptor-specific ligands can therefore be chosen for HCC targeting during the proliferation stage (Figure 2C). On the other hand, heparan sulfate receptors are responsible for the endocytosis of cationic cell-penetrating peptides [47]. Receptors overexpressed on the Kupffer cells (Figure 2D) include galactose, mannose, fucose, and scavenger receptors, while the Fc receptor helps the humoral immune system. Carbohydrate-based materials, such as mannosylated or galactosylated nanoparticles, are more attractive options explored for specific targeting of the Kupffer cells [48]. Apart from collagen type IV and mannose 6-phosphate receptors, retinol binding proteins and platelet-derived growth factors present on the HSCs (Figure 2E) have also been considered for nanoparticle-based drug delivery and have been reviewed separately in the works of Li et al. [49] and Roberts et al. [50]. Sinusoidal endothelial cells (Figure 2F) in the liver have been recognized as major sites for liver metastasis and have recently constituted an interesting lead for immune-modulation [51]. Specific receptors present of the sinusoidal endothelial cells include hyaluronan receptors, low-density lipoprotein receptors, and scavenger receptors. In addition to the receptors expressed on normal liver cells, proteomic results continue to unravel specific epigenetic markers, such as *ADAMTSL5*, *BRD4,* and *H3K27*ac, which can be considered for targeting HCC cells in the future [52,53].

**Figure 2 pharmaceutics-15-01611-f002:**
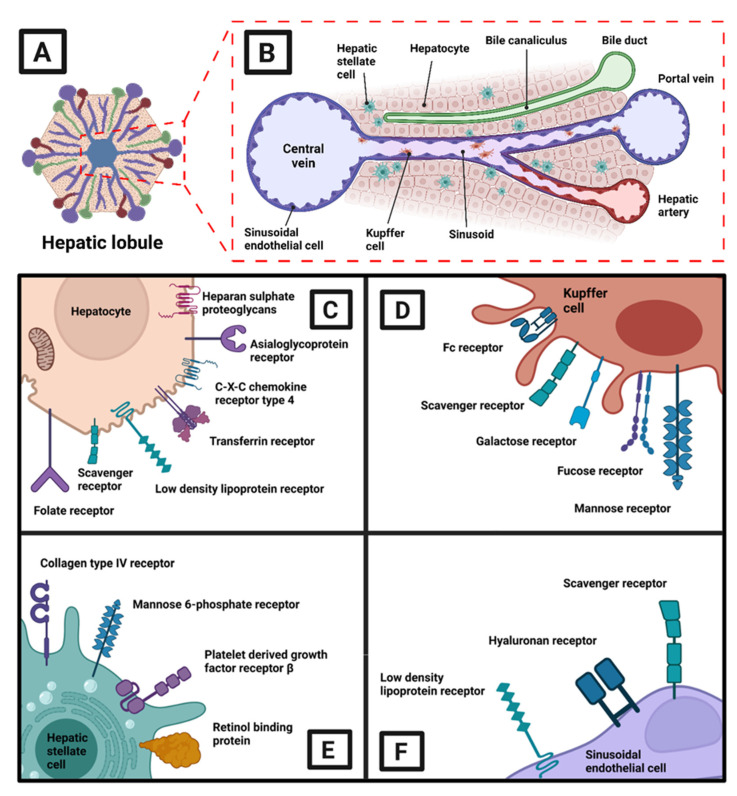
Receptors used for HCC-targeted delivery of nanocarriers on various liver cell types: (**A**) a hepatic lobule; (**B**) the microenvironment of the hepatic lobule; (**C**,**D**) target receptors on hepatocytes (**C**), Kupffer cells (**D**), hepatic stellate cells (**E**), and sinusoidal endothelial cells (**F**), respectively [49]. Understanding the expression of these receptors on specific liver cells can help in the design and development of nanocarriers that can selectively target and deliver drug bioactives for effective treatment of HCC. Figure was adapted from Böttger R. et al. [49] and created with BioRender.com.

**Figure 3 pharmaceutics-15-01611-f003:**
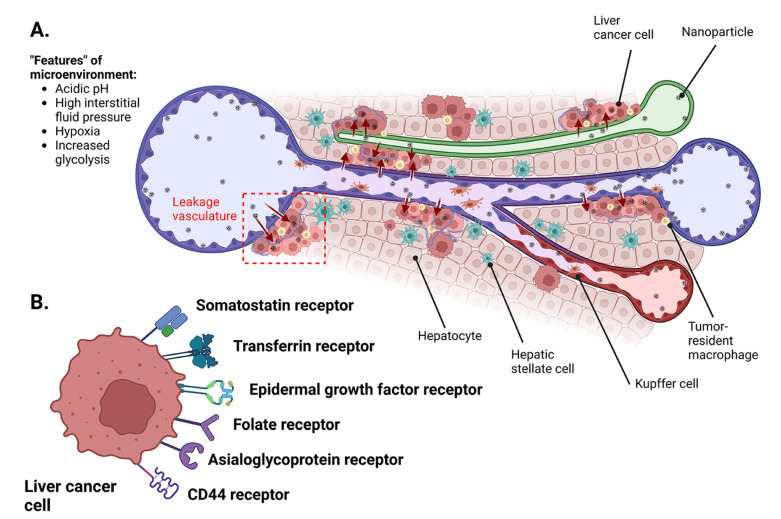
Tumor microenvironment in hepatocellular carcinoma and potential targets for treatment of nanocarriers. (**A**) Passive targeting allows passage of nanoparticles across leaky vasculature and subsequent accumulation in tumor tissues. (**B**) Different receptors on HCC can be actively targeted by surface-functionalized nanocarriers. Figure was created with BioRender.com.

In general, directing nanoparticles toward target tissues can be achieved by passive and active targeting strategies (Figure 3). While the concepts of active and passive targeting of nanoparticles to the liver have been discussed in many works [49,54,55,56], we here sincerely focus on the specific aspects of HCC tumors. Passive targeting is based on the accumulation of nanoparticles in the tumor to attain higher local drug concentrations than in the organs. The efficacy of passive targeting typically depends on the physicochemical properties (size, shape, and surface charge) of the carriers, the route of administration, and more importantly, the EPR phenomenon (Figure 3A). As described in the preceding section, the HCC tumor microenvironment is heterogeneous and hence influences tumor penetration, retention, and extravasation of the nanoparticles. All these variables have been explored to facilitate tumor targeting. For example, nanoparticle extravasation can be reduced by lowering the particle size or modulating blood vessel leakage [57,58]. Innovations in ultrasound-assisted nanoparticle guidance can also promote extravasation. Other factors include nanosurface coatings (such as polyethylene glycol), low interstitial pressure, and ruptured stroma can also enhance tumor entry [59,60]. Active targeting increases the specificity of cargo delivery through functionalization of the nanoparticles to bind to the complimentary receptors (listed in Figure 2B) on HCC cells. Since we addressed the active targeting of normal liver cells in the previous paragraph, here we provide the details regarding somatostatin, epidermal growth factors, and CD44, which are overexpressed primarily on HCC cells. Somatostatin-binding receptors of different subtypes (SSTR1-5) are present in substantial fractions of HCC tumors, and using somatostatin analogs, such as octreotide, can facilitate active targeting of nanoparticles [61]. Epidermal growth factor was first discovered to be involved in HCC development by Kajiyama et al. [62] and has been explored as a target of interest in several works [63,64]. CD44 is well known as a biomarker for cancer stem cell populations and is associated with cancerous pathways, including proliferation and apoptosis [65,66]. Abundant molecules have been investigated for tumor targeting, including antibodies [67], ligands [68,69], aptamers [70], sugars [71], peptides [72], and other moieties. It should however be understood that, upon entering the bloodstream through intravenous injection, all nanoparticles can interact with the serum proteins in the immediate vicinity, leading to aggregation and formation of nanoparticle-protein coronae. The nanoparticle-protein corona is a dynamic layer of protein and other macromolecules that are adsorbed on the nanosurface, and it considerably affects the targeting ability, cellular uptake, and subsequent fate of nanoparticles in vivo [73].

The HCC tumor microenvironment also demonstrates peculiar characteristics, such as hypoxia, extracellular acidosis, and high interstitial pressure, due to a drastic increase in tumorous metabolic activities (Figure 3). Hypoxia can cause resistance to conventional therapy through multiple channels, including apoptosis, drug efflux, and mitochondrial activity, and can interfere with DNA repair mechanisms. Similarly, protons released in the extracellular acidic environment amplify the risk of metastasis and create resistance to various cancer chemotherapeutics [74]. Thus, clinical practice results suggest that a single treatment is not always sufficient to eliminate the entire tumor and prevent cancer metastasis because the tumor microenvironment contains heterogeneous cell populations, many of which can be resistant to single therapies. Recently, multimodal therapy, which incorporates different treatments into a single nanoplatform to yield a stronger therapeutic response, has been proposed to address the impediments associated with single therapies [75]. Physical modalities, such as photodynamic, photothermal, and hyperthermia treatments, allow nanoparticles to generate cytotoxic conditions for tumor ablation. Integration of these technologies into the conventional approaches of chemotherapy to design multimodal therapy could possibly increase efficacy and safety greatly for patients.

## 3. Molecular Mechanisms of Plant Bioactives

It was observed earlier that plant bioactives can inhibit cancer cells through induction of cell differentiation, stimulation of the immune system, nitrosation and nitration suppression, steroidal hormone metabolism, and prevention of DNA binding. More recent reports have explained that these compounds exert their anticancer effects through a variety of cell signaling pathways at multiple levels, such as post-translation regulations, protein synthesis, and intracellular messaging [76]. Multi-OMICS approaches, including genomics, transcriptomics, proteomics, and metabolomics studies, have also revealed intricate anti-HCC mechanisms of plant bioactives in different dimensions that are not usually followed by conventional chemotherapy [25]. Since most of the bioactives exert their anticancer effects via multiple superimposed pathways, the study of their individual activities is not always simple. Identification of major molecular targets and anticancer mechanisms of plant bioactives would enhance the possibility of translational applications in HCC therapy. Various approaches using in vitro experiments and computational tools can help to select the best candidates for the drug-discovery process. In this section we address the various mechanisms of anticancer activities (Figure 4) in HCC cell cultures.

### 3.1. Apoptosis Induction

Apoptosis, or programmed cell death, is an essential cell-death mechanism to maintain cellular homeostasis, and it can be induced either through surface death receptors or via mitochondria-mediated pathways. Cancer cells experience genetic mutations to evade apoptosis and survive under pathological stimuli. Alterations of B-cell lymphoma 2 (Bcl-2) proteins, apoptosis protein inhibitors, death receptors, and executioner caspases are certain features of cancer cells [131]. Targeting these genes and their associated pathways to induce apoptotic cell death (Figure 4) is one of the major anticancer mechanisms of plant bioactives, and study of their molecular interactions have been the focus of many in-vitro and in-vivo investigations. Polyphenols, such as quercetin, epigallocatechin-3-gallate, and carnosic acid, can downregulate more than one anti-apoptotic protein in the B-cell lymphoma 2 family while upregulating the pro-apoptotic ones (e.g., Bax and Bad) in the HepG2 and Huh-7 cells [94,132,133].

NF-κB is a DNA binding protein involved in the pathogenesis of HCC. It is worth mentioning here that NF-κB-related signaling pathways influence cancer-related inflammation, neoplasia, hyperplasia, metastasis, and even chemoresistance [134,135]. Proteins that control apoptosis, such as FLICE-like inhibitory proteins and cellular apoptosis inhibitor proteins 1/2, are stimulated by NF-κB. Activation of NF-κB also cause release of inflammatory cytokines that modulate the tumor microenvironment. Andrographolide, a labdane diterpenoid obtained from *Andrographis paniculata* and its analogs can inhibit NF-κB-related signaling pathways and subsequently modulate p53-induced caspase-3-mediated pro-apoptotic signaling [78]. NF-κB can often enhance its carcinogenic ability by activating the canonical Wnt/β-catenin signaling pathway. A neem terpenoid, nimbolide, has demonstrated the potential to simultaneously inhibit canonical NF-κB and Wnt signaling pathways through downregulation of B-cell lymphoma 2 proteins [136]. Phosphorylation of the NF-κB subunit of p65 could enhance NF-κB transcriptional potential. Isolie, an alkaloid obtained from the embryos of *Nelumbo nucifera,* has been proved to inhibit p65 phosphorylation and induce apoptosis in HCC cells [137]. Although compounds such as 6-gingerol, curcumin, limonene, and betulin (summarized in Table 1) have a similar mechanism in other types of cancers, their effects on p65 phosphorylation have yet to be clarified in HCC cells [134].

### 3.2. Oncogene Inhibition and Tumor-Suppression Gene Expression

Oncogenes and tumor suppression genes primarily regulate the growth of cancer cells. Oncogenes are generally involved in cell division, and their overexpression transforms a normal cell into a malignant one. Overexpression of oncogenes via point mutation or gene amplification is the result of hepatotoxic stress or viral infections, and it facilitates the survival of cancer cells by inhibiting apoptosis (Figure 4). Hence, inhibiting the activity of oncogenes (e.g., transcription factor 1 (*E2F-1)*, pituitary transforming gene 1 (*PPTG1*)) or their upstream genes has been considered a novel therapeutic strategy against HCC. In contrast, certain tumor-suppression genes and proteins (e.g., Kruppel-like factor 6 (*KLF6*), activating transcription factor 3 (*ATF3*), and cyclin-dependent kinase inhibitor protein (p21)) are subdued through abnormal epigenetic pathways, including histone modifications, promoter DNA methylation, and miRNA-mediated post-transcriptional alterations [138]. These epigenetic aberrations reduce tumor suppression activities and increase cancer cell proliferation. Studies with tea polyphenols, such as (-)-epicatechin, (-)-epigallocatechin, and (-)-epigallocatechin-3-gallate, have demonstrated that these compounds form hydrogen bonds with the catalytic pockets of DNA methyltransferase and cause significant inhibition of the enzyme activity, with IC_50_ values ranging from 210 to 470 nM [139]. The alkaloid berberine has also inhibited the proliferation of HCC cancer cells (including Hep3B, HepG2, and SNU-182) via modulation of specific tumorigenesis-related genes for protein expression [138]. The molecular mechanisms of DNA methylation in HCC have been sufficiently described by Liu et al. [140], and reactivation of tumor suppression genes using plant bioactives is considered an emerging epigenetic strategy for HCC treatment. Other recent bioactives, such as Ziyuglycoside II, belonging to this group are included in Table 1.

### 3.3. Cell Cycle Arrest

A typical cell cycle comprises various phases: G1, G2, S, and M. Checkpoints between the phases ensure that DNA replication and cell division proceed accurately, thereby modulating the cell cycle (Figure 4). The cell cycle is controlled by a set of cyclin-dependent kinases (CDKs), which are very specific to each phase of the cell cycle. The activities of the CDKs are further manipulated by the interactions between their subunits, cyclins, and cyclin-dependent kinase inhibitors (CDKIs). While CDK4-cyclin D is responsible for G1 phase progression, the phase transitions from G1 to S and from G2 to M are controlled by CDK2-cyclin E and CDK1-cyclin B, respectively. The use of plant bioactives to modulate cell cycle progression through various mitogenic signaling pathways is currently another attractive venue for cancer intervention. Major active compounds in this category include curcumin, a polyphenol obtained from turmeric; silymarin, a flavonoid from milk thistle; genistein, an isoflavonoid present in soy; mangostanaxanthone, a xanthone isolated from mangosteen; and resveratrol and other polyphenols from grapes. The mechanisms of their anticancer effects have been linked to their interactions with mitogenic signals, such as the activation of mitogen-activated protein kinases and growth factor-receptor interactions, and also modulation in cell survival signals, such as activation of nuclear factor kappa B (NF-κB) [141,142,143] (Figure 4). Independent studies updated in Table 1 have revealed that phytochemicals, such as epigallocatechin-3-gallate, resveratrol, fisetin, and mangostanaxanthone V, can arrest the specific phase transitions in HepG2, as well as in Hep3B cell lines [144].

### 3.4. Antioxidant Effects

Metabolic activities in cell organelles, such as mitochondria, endoplasmic reticulum, and peroxisomes, result in generation of reactive oxygen species (ROS) at picomolar levels. Any amount of excess ROS, such as hydrogen peroxide (H_2_O_2_), superoxide anions (O_2_^−^), singlet oxygen, and hydroxyl radicals (OH^.^), is gradually eliminated by the body. Imbalance in the elimination process leads to accumulation of free radicals, subsequently harming cellular lipids and proteins and causing DNA mutations. Phytochemicals possessing multiple hydroxyl groups, especially the flavonoids, can reduce oxidative stress, scavenge ROS, and thus impede the progression of HCC (Figure 4). It is postulated that flavonoids present a low reduction potential ranging between 0.23 and 0.75 V and exert their antioxidant activity via two one-electron transfer reactions. Flavonoids such as quercetin, kaempferol, naringin, and others are known to inhibit carcinogenesis through similar mechanisms (Table 1), some of which were extensively reviewed by Eid et al. [145].

### 3.5. Anti-Angiogenesis

Angiogenesis is a complex process implicating cell proliferation, migration, invasion, and epithelial mesenchymal transition (Figure 4, Table 1), all of which play key roles in tumor growth and cancer pathology. Apart from tumor growth, angiogenesis is involved in the transformation of dormant micrometastases into clinical detectable lesions [146]. Studies over the last decade have established that miRNAs modulate many of the genes responsible for tumor angiogenesis, including vascular endothelial growth factor (VEGF). The details of the miRNAs and their influences on tumor metabolic and angiogenic pathways were recently summarized by Varghese et al. [147]. Expression of VEGF is the result of tumor response to hypoxic stress and forms the central axis of angiogenesis [148]. Hence, factors inducing VEGF expression and downstream signals following VEGF expression are the potential targets of antiangiogenic therapy. Certain plant bioactives belonging to the alkaloids, terpenoids, flavonoids, and polyphenols have demonstrated anticancer effects through VEGF modulation in the signaling pathways. Alkaloids such as brucine, matrine, and evodiamine impede angiogenesis by targeting the VEGF/AKT/NF-κB pathways. Flavonoids such as resveratrol and cardamonin suppress VEGF signaling, preferably through downregulation of different miRNAs. Reports have indicated that miRNA-21, miRNA-23a, miRNA-16, and miRNA-132 were downregulated following cardamonin treatment. In contrast, resveratrol regulated the expression of miRNA-34a, miRNA-155, miRNA-424, and miRNA-503 [147]. Certain compounds, such as silibinin, curcumin, genistein, and epigallocatechin-3-gallate, have also been effective against HCC tumor angiogenesis [106,149,150]. It should be noted here that the bioactives might not interact through similar pathways or target the same receptor proteins, although they (the bioactives) belong to the same chemical category.

### 3.6. Interference in Cell Signaling Pathways

Cell signaling is the communication network that monitors all cellular activities, such as cell cycle arrest, proliferation, apoptosis, and migration. It is well established that cell signaling pathways are disturbed during HCC, leading to uncontrolled cell division and metastasis. A number of recent studies have demonstrated that targeting cancer-specific pathways using plant bioactives could be a novel therapeutic strategy against HCC (Figure 4). Their molecular targets in cell signaling pathways include kinases, cell membrane receptors, transcriptional factors, miRNAs, cyclins, and caspases [76].

Most of the studies suggested that plant bioactives intervene in certain signaling pathways and regulate the downstream genes/proteins related to cellular functions. Crocin, a carotenoid obtained from saffron, causes inhibition of IL-6/STAT3 pathways and thus promotes cancer cell death via apoptosis [95]. Flavonols from tea extracts demonstrated anticancer effects through modulation of Wnt/β-catenin pathways and their associated genes [151]. Alkaloids such as evodiamine, s-allylmercaptocysteine, and capsaicin have also successfully inhibited the progression of HCC through a similar pathway [91,152,153]. These mechanisms are, however, concentration-dependent, and the effects have yet to be translated to animal models. Other recent bioactives, such as artemisinin, glabridin, luteolin, and pterostilbene, which are associated with cell signaling interactions, are summarized in Table 1.

## 4. Current Nanoparticle-Based Delivery Systems for Plant Bioactives in HCC Therapy

Many preclinical and clinical studies long ago established the anticancer effects of plant bioactives. Polyphenols, terpenoids, alkaloids, phenolic acids, and flavonoids have all exhibited therapeutic potential against HCC cells. Despite such encouraging activities through different pathways, there exist certain biopharmaceutical constraints in the clinical translation of these effects. The diverse functional groups, molecular weights, and polarities of compounds cause wide variations in solubilities and chemical stabilities. The physicochemical properties of plant bioactive are a critical concern in the preformulation processes of phytopharmaceutical formulation. The physical structure and particle size of the bioactives are also involved in their solubility characteristics. Triterpenoids with anti-HCC activity include betulinic acid, ursolic acid, and oleanolic acid, which have polycyclic structures constituted by C_5_ isoprene units, and they consequently exhibited unsatisfactory solubility in physiological media [154]. Conversely, flavonoids and polyphenols possess excess hydroxyl groups that are easily glucuronidated and sulfated in intestinal cells. Subsequently, they are effluxed and result in poor and erratic oral absorption. The presence of unsaturated double bonds renders the flavonoids sensitive to different environmental conditions (temperature, light, oxygen, humidity) and cause interactions with enzymes in different tissues [155], reducing the amount of bioactive reaching the target cancer cells. Bioactives containing alkaloid groups have demonstrated potential as cancer chemotherapeutics. Camptothecin, for example, is a well-known topoisomerase I inhibitor that exhibits low chemical stability and renal toxicity [156]. Another alkaloid, berberine, was reported for its very low bioavailability (~5%) due to its affinity for plasma proteins [157]. Consequently, many of the plant bioactives have been classified as biopharmaceutical classification system (BCS) class IV drugs, which have low aqueous solubility, low gastrointestinal permeability, and rapid elimination.

Considerable efforts have been devoted to the design of nanoparticle-based delivery systems since they can overcome critical problems, such as chemical instability, poor solubility, low bioavailability, and drug-resistance, that are often associated with conventional phytomedicines. Furthermore, the nanoencapsulation process increases the shelf-life of bioactives with controlled release opportunities at the target site. Multifunctional nanoparticles with tunable surface chemistry can navigate through the defective vascular structure of tumors. They exhibit great potential to achieve accurate treatment through cell-specific targeting and transporting payloads to specific organelles [158]. With the increasing number of cancer cell receptors now being identified, complimentary peptides have been attached to the nanocarrier surface to attempt to specifically deliver active compounds to target cancer cells. This strategy was mostly based on receptor-ligand-mediated endocytosis aiming to reduce systemic toxicity and decrease drug resistance [159]. The tumor microenvironment characteristics are also systematically exploited for designing bioresponsive carriers capable of on-demand drug release. Changes in pH level, hypoxia, hyperthermia, redox potential, and expression of specific enzymes in the HCC tumor are being explored as internal stimuli to trigger drug release selectively inside cancer cells. External stimuli (such as light, heat, ultrasound, and magnetic interferences) are combined with or without chemical conjugation of internal stimuli to increase anticancer activity [74]. Well-designed bioresponsive nanoparticles are supposed to dynamically detect the HCC tumor microenvironment and thereby avoid premature release of the bioactive compounds. Extensive theories and reviews of the use of variety of specific stimuli for nanoparticle drug release can be found in the recent literature [159,160,161,162].

With the continuous progress in the field of cancer therapy, numerous approaches with nanomedicines proposed to fight HCC have been reported. However, only a handful have been approved for clinical trials. To improve the clinical translation of nanomedicines, the United States National Science and Technology Council launched the National Nanotechnology Initiative in 2000 [163]. This section describes the clinically approved nanoparticles (Figure 5) that have been administered in cancer treatments and summarizes the various strategies for the development of phytochemical-based nanomedicines for HCC. This knowledge would increase familiarity with the ongoing trends in the development of nanomedicines, particularly for clinical trials.

### 4.1. Liposomes and Their Derivatives

Liposomes are classic nanoparticulated-based delivery systems with a long track record of successful transition as anticancer therapeutic carriers. Liposomes are generally composed of phospholipids, and they assume a unilamellar or multilamellar vesicular structure in a size range of 50–200 nm (Figure 5, Table 2). Due to the presence of phospholipids, liposomes are highly biocompatible and can deliver both hydrophilic and lipophilic drugs to target cells. Recent advancements have significantly overcome the stability problems associated with liposomes through modulation of surface charges, size, lipid composition, and surface functionalization using targeting ligands [164,165]. Surface modification also allows for site-specific delivery of bioactives and prolongs their systemic circulation. Encapsulation of resveratrol in cationic liposomes improved its systemic bioavailability by 3.2 folds and increased the localization of the drug in cancerous tissues. The formulation was further tested for its therapeutic and preventive efficacy in HCC-induced rats, and the results were well collaborated [166]. Liposomes are also convenient for encapsulation of crude plant extracts. It is worth mentioning here that certain crude extracts have been found to exhibit the ‘entourage effect,’ in which the crude product demonstrated higher anticancer activity than that of the individual purified compounds. Yue et al. showed that liposomes containing *Brucea javanica* oil extract induced apoptosis in HepG2 cells in a dose-dependent manner (from 2.5 mg/L to 5 mg/L). Upon administration in mice, the liposomes caused a marked change in tumor pathology through apoptosis, which was established using TUNNEL staining [167]. Other forms of liposomes, such as nanoliposomes and niosomes, have also been utilized for the delivery of both crude extracts and isolated compounds to HCC cells (Table 2). While niosomes have been preferred to conventional liposomes due to their improved stability, nanoliposomes offer cell-targeting abilities. For example, Tian et al. recently surface modified nanoliposomes with hyaluronic acid and glycyrrhetinic acid for the delivery of curcumin to HCC cells and hepatic stellate cells at the same time; the nanoliposome could significantly inhibit metastasis and initiate tumor microenvironment remodeling [168]. Such strategies have been demonstrated to be useful, especially for targeting a heterogeneous population of cancer cells that tend to cause drug resistance.

### 4.2. Solid Lipid Nanoparticles

The second generation of lipid-based nanocarriers, which are available in the clinical scenario, more recently in form of COVID-19 vaccines, are solid lipid nanoparticles [207]. Solid lipid nanoparticles are colloidal particles within a range of 50–200 nm and made of biodegradable lipids, such as fatty acids, triglycerides, and waxes (Figure 5, Table 2). These nanoparticles can be prepared through a variety of processes, including high-pressure homogenization, ultrasonication, solvent evaporation, microemulsion methods, and microfluidization. Like liposomes, solid lipid nanoparticles can demonstrate certain benefits, including encapsulation of both hydrophilic and lipophilic molecules, low toxicity, and high bioavailability. Nonetheless, their rigid core improves their stability compared to liposomes. These nanoparticles have been perceived to overcome several physiological barriers that impede bioactive delivery to tumor tissues and can also evade multidrug resistance mechanisms [208]. For example, resveratrol was loaded into cationic solid lipid nanoparticles that were designed to improve cellular receptor interaction and cellular uptake [194]. This result was perhaps due to nanoparticle interactions with the anionic cellular components, such as glycoproteins, proteoglycans, and phosphatidylserine, present on the cancer cell surface. Quercetin is another bioactive that revealed higher cell penetration capacity upon being loaded into phytosterol-containing solid lipid nanoparticles [193]. Similar to the previous work, this research was another attempt to increase cellular uptake through fluidization of bilayered cell membranes using sterols. Interestingly, many seed oils possess remarkable anticancer properties and can conveniently be encapsulated within solid lipid nanoparticles. Nanoencapsulation of coix seed oil revealed an increase in cytotoxic effects by 1.52 to 3.24 folds, although the release of the cargo was not favorable at low “cancerous pH” levels [192]. Lipid nanoparticles have been exploited as platforms for combination therapy. Zhao et al. developed lipid nanoparticles for the co-delivery of doxorubicin and curcumin against HCC and found a synergistic interaction between the compounds [190]. While a microfluidizer was used to synthesize monodisperse lipid nanoparticles with 80-nm diameter, the study was also an excellent example emphasizing the combined effects of chemotherapeutic agents and chemosensitizers on cancer progression through apoptosis, proliferation, and angiogenesis signaling pathways.

### 4.3. Polymer-Based Nanoparticles

Polymer-based nanoparticles are prepared using natural, synthetic, or semi-synthetic materials, which allow for easy manipulation of size (25–500 nm), shape, and surface properties (Figure 5, Table 2). Polymers such as chitosan, poly(lactic-*co*-glycolic acid) (PLGA), and polyethylene glycol, have been commonly used for preparation of both nanocapsules (cavities enclosed within polymer shells) and nanospheres (matrix shape). Nanoprecipitation, emulsification, ionic gelation, and microfluidics are some of the preferred methods for the synthesis of polymeric nanoparticles. Bioactive compounds can be entrapped within the polymer matrix, encapsulated inside the particle core, physically bound to the nanosurface, or chemically grafted into the polymer backbone. Polymeric nanoparticles can, therefore, deliver a wide range of compounds with different polarities and molecular weights. In addition, drug loading and their release kinetics can be tailored through modulation of bulk composition and surface functionalities. Phytochemicals such as umbelliferone β-D-galactopyranoside, apigenin, quercetin, gallic acid, and ellagic acid have demonstrated promising effects against HCC in various cell lines, as well as in vivo studies (Table 2). The release of these phytochemicals from the polymer matrix were studied in various conditions, including neutral and acidic pH levels, as well as in the presence of 10 mM glutathione [196,200,201]. It should be noted that the molecular characteristics, especially the orientation of various ionization groups in the compounds, must be well studied prior to the selection of the polymer or polymer-modification strategies to achieve the desired release. The surface charge density of nanoparticles influences the loading and release profiles of plant bioactives; positively surface charged entities have been found to be more susceptible to cellular uptake than anionic or nonionic (e.g., PEGylated) nanoparticles. However, a very strong, positive, surface-charged density can cause cell toxicity by damaging cell membranes [209].

### 4.4. Metallic-Based Nanoparticles

Metallic-based nanoparticles, known as a theranostic platform, have been exclusively explored in preclinical and clinical trials for the detection and treatment of certain diseases (Figure 5). FDA-approved metallic-based nanoparticles have demonstrated the ability to deliver active molecules to the targeted cells, thereby reducing their side effects. Most of the metallic-based nanoparticles approved for cancer therapy lie in the range of 5–50 nm and decompose under physiological conditions through various metabolic pathways and do not damage healthy tissues [210]. Among nanoparticles, gold nanoparticles have received widespread attention due to their biocompatibility, optical properties, and physicochemical properties, which are impossible with organic particles. Gold nanoparticles can also be easily functionalized with a variety of targeting molecules or plant bioactives, thus imparting additional properties and delivery capacities. Although there are numerous approaches for the synthesis of gold nanoparticles using plant extracts, reports on gold nanoparticle-assisted delivery of plant bioactives in HCC are limited. A study conducted by Krishnan et al. showed that hesperetin-conjugated gold nanoparticles suppressed mast cells, TNF-alpha, and NF-κB in diethylnitrosamine-induced hepatocarcinogenic conditions [204]. However, the researchers failed to reveal how nanogold affected the apoptotic activity of hesperetin [211]. In contrast, it was demonstrated by Zhang et al. that flavonoids, such as resveratrol, tethered to nanogold can exert superior anti-HCC effects compared to the free compounds [205]. Although other major plant bioactives, including epigallocatechin-3-gallate and kaempferol, have also been conjugated with gold nanoparticles and tested as anticancer therapeutics, their effects specifically on HCC cell lines or xenograft models have yet to be disclosed [212].

## 5. Current Challenges and Future Perspectives

Nanotechnology has provided the opportunity to increase the stability and targetability of plant bioactive-based nanocarrier systems. Although the anti-HCC efficacies of plant bioactive-based nanoparticles have been tested mostly in cell cultures and chemically-induced (such as diethylnitrosamine) tumor models, their therapeutic activities may not correlate with the therapeutic efficacies in the complex HCC pathophysiology in vivo. Advanced studies in the management of metastasis could also be conducted, especially for plant antioxidant compounds that can modulate ROS generation [213,214]. Therefore, in vivo models that are closely related to clinical settings are required to evaluate the potential candidacy of plant bioactives.

The therapeutic activity of curcumin has been well established since it could block the P-glycoprotein-1 pump that is responsible for the development of multi-drug resistance against the classic anticancer drugs, such as paclitaxel, 5-fluorouracil, and doxorubicin. Other bioactives, such as apigenin, baicalein, and quercetin, have also been investigated in combination with these classic drugs for the modulation of P-glycoprotein and pro-apoptotic activity [215,216,217]. Nevertheless, due to biopharmaceutical constraints, such as poor solubility and erratic systemic absorption of these molecules, most of the reports have been based on cell cultures or sometimes computational studies [218,219,220]. Similar trends have been observed with other hydrophobic bioactives, resulting in difficulty with drug screening and preformulation processes. Liposomes with targeted ligands have been commonly explored as suitable carriers for codelivery of both hydrophobic and hydrophilic molecules [221] and could play a significant role in cotransport of both classical drugs and plant bioactives. The next generation of nanoparticle-based delivery systems, including solid lipid nanoparticles and polymer-based nanoparticles, could improve the stability and encapsulation efficacy of bioactive compounds. At the same time, the effect of plant bioactives on drug resistance and multi-drug resistance should be investigated. In fact, only a few recent works have revealed that codelivery of plant bioactives and synthetic drugs can significantly reverse drug resistance in animals [222]. The anti-HCC blockbuster sorafenib has been tested in combination with flavonoids such as rutin, fisetin, and silibinin to overcome chemoresistance, as well as to alleviate drug-related toxicity in various kinds of cancer models [223,224,225]. Similarly, wortmannin, a compound isolated from the fungus *Penicillium funiculosum*, has exhibited reverse platinum resistance in ovarian cancer when codelivered with cisplatin in nanoform [226]. Such combination therapies could increase the possibility for new chemo-sensitizers, which can be explored to overcome multidrug resistance associated with HCC therapy. Combinations of epigallocatechin-3-gallate and theaflavin coloaded into PLGA nanoparticles have been found to enhance the anticancer effects of cisplatin in lung carcinoma, cervical carcinoma, and leukemia [227]. Based on the proportions of the compounds used, the drug combinations indicated synergistic effects; thus, the further study of drug-resistance mechanisms is required for dose and toxicity evaluations [228].

Theranostic nanomedicines integrate diagnostics and therapeutic functions into one platform that can monitor the accumulation of therapeutics, as well as disease progression. Preliminary works using simple polyphenols, such as curcumin, gallic acid, and quercetin have been encouraging [229,230,231]. Alternatively, FDA approved polymers, such as PLGA, have been used in combination with other photoluminescent materials and iron nanoparticles to develop the theranostic features of fluorescent imaging and magnetic resonance imaging. This type of concept could provide additional information for real-time monitoring, pharmacodynamics, and pharmacokinetics in vivo, which could be very useful for optimizing drug loading and carrier designs [232].

Colloidal gold has been extensively studied for metallic-based theranostics in various in vitro and in vivo models, occasionally reaching clinical trials. Several formats of gold nanoparticles, such as gold nanorods, nanostars, nanocages, and nanocages, have demonstrated significant effects due to their excellent optical and physical properties [233]. They could be further combined with plant bioactives and gold/biopolymers with FDA approval as nanotheranostic systems in clinical studies. At the molecular level, gold nanomaterials can affect HCC signaling pathways and hinder cancerous activity, such as self-renewal and cell differentiation. Novel platforms of gold and plant bioactives, along with an appropriate targeting ligand, could present a powerful synergistic tool to modulate cell-signaling pathways and deter tumor progression. Long-term toxicity studies of such polymer and metallic-based nanotheranostics must be performed for the safety of patients.

Nonspecific interactions with blood cells and serum protein after in vivo injection are a major drawback of nanocarriers, and their toxicity has been reported in some animal models [234]. Nanotoxicity is indeed a big challenge and is dependent on various factors, such as biological environment, physicochemical property, interactions between drugs and nanocarriers, and interactions between nanocarriers and cells/extracellular matrix. Each nanomedicine has its issues, making evaluation of chronic and acute toxicity of the nanomaterials more difficult in clinical trials. In fact, there is no standard list of required biocompatibility assays, even after several interventions by the FDA and European Medicines Agency (EMA, Amsterdam, The Netherlands) [235]. With the advancement of biomimetic technology, cancer cell membrane-based nanoparticles have become an interesting area in cancer therapy. These nanoparticles can evade immune recognition and exhibit homologous adhesion ability due to their unique membrane proteins. Furthermore, they have demonstrated a certain capacity to improve the tumor microenvironment using a biomimetic nanoplatform [236]. Therefore, cancer cell-based nanoparticles have received extensive attention due to their novel carrier features for phytochemical compounds. New-generation delivery shuttles, such as exosomes, have been recently tested for the encapsulation of plant bioactives [237,238]. They are a different class of biomimetic carriers and can be designed to perform theranostic applications [239]. As mentioned above, it should also be realized that nanoparticle interactions with acellular materials, such as extracellular matrix, need to be exclusively explored to critically understand nanoparticle penetration and drug penetration within the tumor, and only few studies have been conducted in this area.

## 6. Conclusions

The occurrence and progression of HCC are complex and involve multistep treatment lowering the five-year survival rate by less than 20% [240]. Limitations in conventional modes of treatment lead to the failure of chemotherapeutic regimens and are related to the development of drug resistance, insufficient efficacy, metastasis, and undesired side effects. There was evidence that demonstrated the potential activity of plant bioactives against HCC during disease progression at molecular levels through the modulation of cell cycles, signaling pathways, and/or gene expression. Additionally, plant bioactives can evidently manage issues of drug resistance when delivered in combination with classic chemotherapeutics.

However, plant bioactives come from different plant species and vary in their physicochemical properties. Consequently, they have different constraints for in vitro and in vivo administrations. Delivery of plant bioactives using nanoparticle-based carrier systems can provide better control in drug release, protect unstable compounds, and subsequently exhibit high bioavailability. We highlighted the key parameters, advantages, and disadvantages of the most employed nanocarrier systems for plant bioactives, including liposomes, solid lipid nanoparticles, polymer-based nanoparticles, and metallic-based nanoparticles. Formulation objectives in terms of stability, efficacy, immunogenicity, and biodegradability must also be considered. While liposomes appear convenient for coencapsulation of both hydrophilic and lipophilic molecules, solid lipid nanoparticles are a better choice in terms of stability. Alternatively, although polymer-based nanoparticles support large volumes of cargo, they might not be biodegradable in all cases. Metallic-based nanoparticles are inferior as phytochemical carriers, but they present theranostic ability. Hence, the selection of the carrier system must complement the efficacy and physicochemical properties of the phytochemical encapsulated.

Designing plant-based nanomedicines is nevertheless complicated and does not always meet the clinical demands because of the diversity of chemical structures of the phytochemicals and their ability to participate in nanoparticle formation. In simpler terms, one common nanoencapsulation process using the same type of polymer-, lipid-, or metal-based nanocarriers may not necessarily be guaranteed to be the best encapsulation process for other phytochemical compounds because the nanoparticles are always fabricated in different environmental conditions in which the interactions between the polymer/lipid and phytochemical compounds are different. Thus, to customize control of each phytochemical compound over its nanoencapsulation process requires in-depth understanding of the molecular configurations of the phytochemicals, its intermolecular interactions, and its synthesis parameters. Tuning of the nanoparticle surface charge is equally important to control their stability and biological activities. Most of the literature has gradually recognized cationic nanocarriers as effective since they facilitate better cellular uptake through interactions with anionic phosphatidylserine residues in HCC tumors. Trying to target HCC cells through prior identified receptors of HCC cells could be another challenge for active targeting because both the underlying cellular heterogeneity and the tumor microenvironment play major roles in disease progression. Critical considerable parameters, such as preformulation study (solubility, partition coefficient, crystallinity, polarity, intrinsic dissolution rate of phytochemicals), reproducibility, stability, and safety need to be considered toward the management of plant bioactives with nanocarrier systems. Nevertheless, better understanding of HCC tumor heterogeneity and the development of state-of-art drug-carriers and plant bioactive-based nanomedicines would undoubtedly provide new opportunities for anti-HCC therapy.

## Figures and Tables

**Figure 1 pharmaceutics-15-01611-f001:**
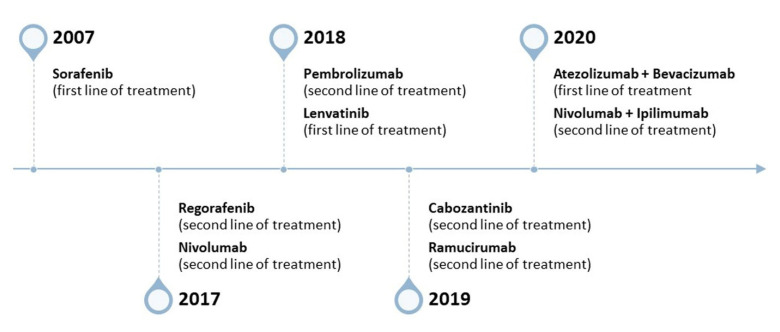
Timeline for FDA-approved drugs for HCC treatment between 2007 and 2020; data from 11–19.

**Figure 4 pharmaceutics-15-01611-f004:**
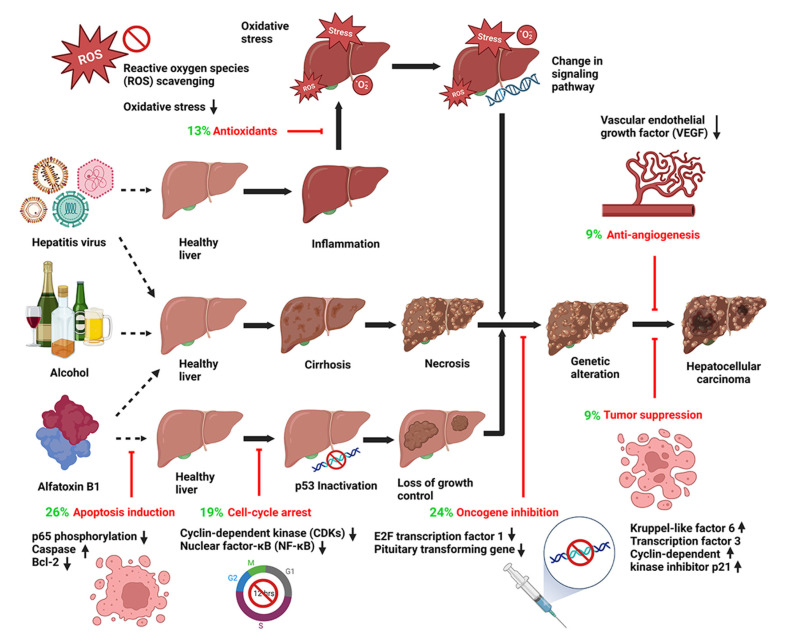
Progression of hepatocellular carcinoma and molecular mechanisms of plant bioactives against various oncogenic pathways. The frequencies of compounds targeting in each pathway have been highlighted in green. The percentage was calculated from data in Table 1 (105–153), and the figure was created with BioRender.com.

**Figure 5 pharmaceutics-15-01611-f005:**
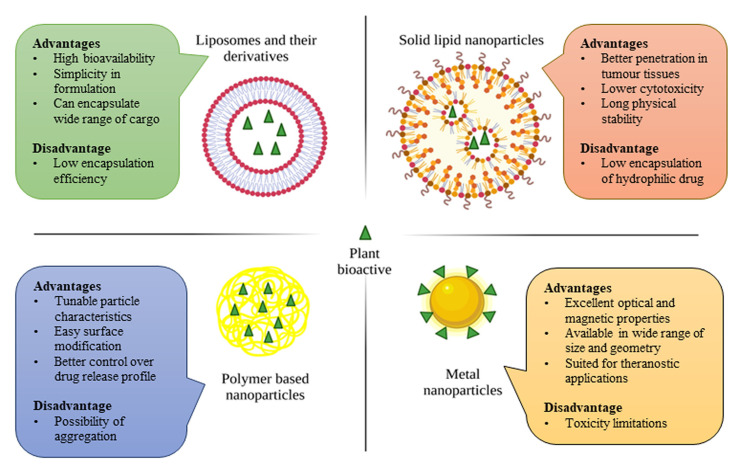
Major categories of nanoparticle-based delivery systems in phytopharmaceutical formulations and their characteristics. Data from 164, 171, 176, and 182, and the figure was created with BioRender.com.

**Table 1 pharmaceutics-15-01611-t001:** Summary of different mechanisms of plant bioactives against HCC and their IC_50_ values as observed in preclinical studies.

Plant Bioactives (Biological Source)	Animal Model/Cell Lines	Mechanism of Action	IC_50_	References
(14E, 18E, 22E, 26E)-methylnonacosa- 14, 18, 22, 26 tetraenoate (*Amaranthus spinosus*)	HepG2	Inhibition of proliferation by upregulation of Bax; downregulation of Bcl-2 and cyclin B, resulting in G2/M arrest	25.52 µM	[77]
Andrographolide (*Andrographis paniculata*)	Cisplatin-resistant HepG2 (HepG2CR)	Sub-G1 phase arrest; apoptosis; antiangiogenesis	40 µM	[78,79]
Astrakurkurone (*Astraeus hygrometricus*)	HepG2, Hep3B	Inhibition of proliferation through cycle arrest at sub-G0/G1 phase; upregulation of pro-apoptotic markers Bax and cleaved caspase 9, with downregulation of antiapoptotic marker Bcl-2	150 µM in HepG2, 40 µM in Hep3B	[80]
Allicin (*Allium sativum*)	HepG2, Hep3B	Autophagic and apoptotic cell death through ROS generation	35 µM in HepG2, 35 µM in Hep3B	[81]
Ardipusilloside I (*Ardisia pusilla*)	HepG2, SMMC-7721	Inhibition of growth, invasion, and metastasis through suppression of MEK/ERK and Akt signaling pathways; inhibition of metastasis through upregulation of E-cadherin	-	[82]
Artemisinin (*Artemisia capillaris*)	SMMC-7721	Inhibition of proliferation through blocking of PI3K/Akt and mTOR signaling channels; induction of apoptosis through downregulating antiapoptotic proteins XIAP and survivin and upregulating proapoptotic proteins cleaved caspase-3 and PARP; impeding metastasis through increasing cell–cell adhesion; inhibiting invasive and migratory ability	-	[83]
Berberine (*Berberis vulgaris*)	HepG2	Decreased proliferation and induced apoptosis through suppression via p65 of NF-kB pathway	3587.9 µM	[84,85]
Betulinic acid (*Betula pubescens*)	HepG2, SMMC-7721.	Causing apoptosis through mitochondrial pathway	24.8 µM in HepG2, 28.9 µM in SMMC-7721.	[86,87]
Boldine (*Peumus boldus*)	HepG2, Wistar rats	Induced apoptosis; overexpression of Bax and cleaved caspase 3	170 µM in HepG2	[88,89]
Caffeine (*Coffea arabica*)	SMMC-7721, Hep3B	Working in combination with 5-fluorouracil to reduce proliferation and induce apoptosis through intracellular ROS production	2.2 mM in SMMC-7721, 2.02 mM in Hep3B	[90]
Capsaicin (*Capsicum annuum*)	HepG2	Improved antitumor effect of sorafenib and induced apoptosis through intracellular ROS production	150 µM	[91,92,93]
Carnosic acid (*Rosmarinus officinalis*)	HepG2, SMMC-7721	Inhibited cell proliferation; induced apoptosis through increased production of ROS	43.7 µM in HepG2, 74.8 µM in SMMC-7721	[94]
Crocin (*Crocus sativus*)	HepG2, HCCLM3 cells	Induced autophagic apoptosis in an Akt/mTOR-dependent mechanism; inhibition of IL-6/STAT3 pathways	-	[95]
Curcumin (*Curcuma longa*)	HepG2, rat model	Modulated TGF-β, AkT, and caspase-3 expression; protective effects against toxins through expression of nuclear factor E2-related factor 2 and glutathione	23.15 µM	[96,97]
Damnacanthal (*Morinda citrifolia*)	HepG2	Decreased the phosphorylation levels of Akt; targets matrix metalloproteinase-2 secretion; induces apoptosis	5.1 µM	[98]
Eriocitrin (*Citrus limon*)	HepG2	Decreased proliferation through cell cycle arrest at G2 phase; induced apoptosis through increased expression of pro-apoptotic proteins Bcl-2, caspase 3, caspase 8, caspase 9, PARP, TNF receptor, NF-κB, and IkB; downregulated antiapoptotic genes.	-	[99]
Epigallocatechin-3- gallate (*Camellia sinensis*)	HepG2, Hep3B, Huh-7	Reduced proliferation through inhibiting ERalpha36 and PI3K/Akt and MAPK/ERK pathways; caused apoptosis by caspase 3 activation and induction of the ER-36-EGFR-Her-2 feedback loop	74.04 µM in HepG2, 50.8 µM in Hep3B, 83.8 µM in Huh-7	[100,101,102]
Emodin (*Rheum palmatum*)	Huh7, Hep3B, HepG2	Cell cycle arrested at G2/M phase	101.5 µM in Huh7, 66.9 µM in Hep3B, 74.36 µM in HepG2	[103]
Fisetin (*Rhus cotinus*)	HepG2	Prevented proliferation through cell cycle arrest; stimulated apoptosis and necroptosis through increased expression of Bax, caspase-3, TNF-alpha, and PARP and through increased expression of RIPK1, RIPK3, pRIPK1, pRIPK3, and MLKL; reduced expression of pNF-κB, NF-κB, and pIKKB	3.2 µM	[104,105]
Genistein (*Millettia reticulata*)	SK-Hep-1, Huh-7, Hep3B	Increased protein expression of Fas, FasL, and p5; impeded tumor growth through cell cycle arrest at G0/G1 and G2/M phases	16.23 µM in SK-Hep-1, 18.67 µM in Huh7	[106,107]
Ginsenoside Rh2 (*Panax ginseng*)	HepG2	Causing apoptosis through mitochondrial pathway.	100 µM	[86,108]
Glabridin (*Glycyrrhiza glabra*)	HepG2, Huh-7, MHCC97H, Sk-Hep-1	Reducing stemness by inhibition of TGF-beta/SMAD2 signaling channel; reduced invasive ability through downregulation of MMP-9 and MMP-1; preventing tumor formation in xenograft model	7.22 µM in HepG2	[109,110]
Kaempferol (*Camellia sinensis*)	Huh-7	Inhibits p44/42 MAPK and hypoxia-inducible factor 1 activity	4.75 µM	[111]
Lanatoside C (*Digitalis lanata*)	Huh-7	Inhibition of proliferation through cell cycle arrest; induction of apoptosis through JNK pathway activation and ROS generation	-	[112]
Luteolin (*Verbascum lychnitis*)	HCC cells from rats	Causing cancer cell death through increased production of ROS and release of cytochrome-c; prevented growth through increased expression of *miR-6809-5p*, blocking activation of growth cell signaling regulator FLOT1	12 µM	[113]
Naringin (*Vitis vinifera*)	HepG2	Upregulates the expression of *miR-19b* mRNA and induces cell apoptosis	20 µM	[114]
Neferine (*Nelumbo nucifera*)	Hep3B	Causes apoptosis through downregulation of cell cycle markers and induction of ER stress	14.8 µM	[115,116]
Oleanolic acid (*Ophiopogon japonicus*)	Huh-7	Induction of apoptosis through increased mitochondrial permeability, causing activation of certain proapoptotic markers; inhibition of expression of XIAP in cancer cells	100 µM	[117]
Oroxylin A (*Oroxylum indicum*)	HepG2	Reduced metabolic ability of cancer cells under hypoxic conditions by inhibiting the generation of lactate and glucose; suppresses expression of metabolic regulator HIF-1a; caused differentiation of cancer cells through activation of HNF-4a, thereby reducing metastatic ability	-	[118]
Protopanaxadiol (*Panax ginseng*)	HepG2, PLC/PRF/5	Inhibition of EMT through higher expression of E-cadherin and reduced expression of vimentin; inhibition of EMT also through restriction of STAT3 activation and through inhibition of Twist1 expression	~70 µM in all cell types	[119]
Pterostilbene (*Pterocarpus marsupium*)	HepG2	Prevented migration, invasion, and proliferation through downregulation of MMP-9 and through suppression of TPA-induced PI3K-Akt-NF-κB signaling; inhibits metastasis	39.06 µM	[120]
Quercetin (*Allium cepa*)	HepG2	Caused apoptosis through upregulation of p53 and Bax; impeded glycolysis through reduction in glycolysis enzyme HK-2 and by reducing expression of phosphorylated mTOR and Akt	24 µM	[121,122,123]
Resveratrol (*Vitis vinifera*)	SMMC-7721, HepG2	Limited cell growth through inhibition of metabolic phenotypes that facilitate anaerobic growth	100 µM in SMMC-7721, 64.5 µM in HepG2	[124,125]
Rutin (*Fagopyrum esculentum*)	HepG2	Inhibition of cell proliferation; inhibited protein expression of cytochrome P450-dependent *CYP3A4*	52.7 µM	[126]
Tatariside F (*Fagopyrum tataricum*)	H22	Caused apoptosis through upregulation of p53 and Bax and down-regulation of Blc-2; inhibits tumor growth in vivo	1.31 µM	[127]
Thymoquinone (*Nigella sativa*)	HepG2, SMMC-7721	Activation of caspases and generation of ROS	84.2 µM in HepG2, 91.6 µM in SMMC-7721.	[128]
Ursolic acid (*Vaccinium macrocarpon*)	HepG2, Huh-7	Inhibition of proliferation through disruption of DNA structures, leading to cell cycle arrest; increased expression of p21/WAF1, inducing cell cycle arrest and apoptosis; inhibition of expression of XIAP in cancer cells	-	[129]
Ziyuglycoside II (*Sanguisorba officinalis*)	HepG2, SMMC-7721	Inhibited cell cycle proliferation and caused apoptosis through cell cycle arrest; suppression of migration and invasion through downregulation of MMP2 and MMP9, while also inhibiting the EGFR/NF-kB pathway	13.1 µM in HepG2, 15.6 µM in SMMC-7721.	[130]

**Table 2 pharmaceutics-15-01611-t002:** Plant bioactive-based nanoparticles against HCC in preclinical studies.

Plant Bioactives	Observations and Outcomes	Cellular/Intracellular Target	References
**Liposomes ** **[169]**			
Aprepitant and curcumin	Reduced ECM deposition and tumor angiogenesis	Drug accumulation in tumor tissues by EPR effect and GA and/or CD44 receptor-medicated endocytosis	[168]
Betulinic acid	Enhanced cell apoptosis and mitochondrial membrane disruption in HepG2 cells	Mitochondrial membrane of HepG2 cells	[170]
*Bistorta amplexicaulis* extract	Plant extract containing nanoliposomes demonstrated higher cytotoxicity toward HepG2 cells	HepG2 cells in vitro	[171]
*Brucea javanica* extract	Increased apoptosis of HepG2 cells	DNA synthesis inhibition and blockage of G0/G1 development to S phase	[167]
Celastrol	Suppressed AKT activation, induced apoptosis, and retarded cell proliferation	Uptake in HepG2 cells in vitro through receptor-mediated endocytosis	[172]
Curcumin	Galactose-morpholine modification resulted in better lysosomal targeting efficacy	ASGPR receptors on liver cells in mice	[173]
Curcumin and cisplatin	Exhibited synergistic effects in mouse hepatoma H22 and human HCC HepG2 xenograft models	Nanoliposomes delivered both curcumin and cisplatin to tumor tissues	[174]
Garcinia	Drug loaded nanolipoprotein complex showed higher cell death rate compared to free drug	Scavenger receptor class B type 1 receptors	[175]
Honokiol	Inhibited tumor metastasis by destabilizing EGFR and reducing the downstream pathways	Cellular uptake study was not performed	[176]
Nitidine chloride	Exhibited sustained release and higher cytotoxicity toward Huh-7 cells	Huh-7 cells in vitro	[177]
Oleanolic acid	Suppressed growth of murine H22 hepatoma and prolonged the survival of tumor-bearing mice	Cellular uptake study was not performed	[178]
Resveratrol	Improved localization of drug in cancer tissue by 3.2 and 2.2 fold increases, respectively, in AUC and Cmax	HepG2 cells in vitro; cancer tissues in rat liver	[166]
Silibinin and glycyrrhizic acid	Synergistic effect of silibinin with glycyrrhizic acid on HepG2 cell line	Cellular uptake study was not performed	[179]
Tanshinone IIA	Promoted apoptosis in HepG2 and Huh-7 cells	Galactose modified niosomes targeted ASGPR receptors on hepatocytes	[180]
Timosaponin AIII and doxorubicin	TAIII improved uptake of doxorubicin HCC cells and exhibited synergistic effect	HepG2 cells in vitro, and tumor bearing mice model	[181]
Triptolide	Induced cell proliferation arrest and apoptosis via the mitochondrial pathway	Huh-7 cells in vitro, and tumor sites in mice model	[182]
Triptolide and Ce6	Under NIR laser irradiation, liposome released triptolide and, along with Ce6, caused apoptosis of HCC cells	HepG2 cells in vitro, and patient-derived tumor xenograft	[183]
Triptolide and sorafenib	Long circulating liposomes promoted cancer cell apoptosis and inhibited tumor growth through synergistic effects	Huh-7 cells in-vitro, and tumor sites in mice model	[184]
Ursolic acid and ginsenoside	Intervened cell proliferation, apoptosis, and cell cycle of HepG2 cells	Cellular uptake study was not performed	[185]
β-sitosterol	Improved cellular uptake and cytotoxicity in HepG2 cells; increased drug-plasma concentrations by 8 fold	HepG2 cells in vitro	[186]
**Solid lipid nanoparticles ** **[187]**			
Cantharidin	Inhibited tumor growth and prolonged survival in tumor-bearing mice	Hyaluronic acid surface functionalization improved nanoparticle uptake in tumor tissues of rats	[188]
Capsaicin	Stable in circulation for a period of three days	Biodistribution studies revealed nanoparticles accumulated in the liver	[189]
Doxorubicin and curcumin	Synergistic activity was observed, including reversal of multidrug resistance	Cellular uptake and biodistribution study was not performed	[190]
Ganoderic acid	Exhibited significant antitumor effect in vivo by balancing hepatic injury markers, biochemicals, and antioxidants markers	Rapid internalization of nanoparticles in HepG2 cells	[191]
Naringin and coix seed oil	Exhibited synergistic effect by enhancing antitumor activity in xenograft model	Cellular uptake study was not performed	[192]
Quercetin	Creating better penetration into HepG2 cells	-	[193]
Resveratrol	Caused reduction in tumor volume and accumulation of drug in tumor tissues	Accumulation of drug in livers of rats	[194]
**Polymer-based nanoparticles ** **[195]**			
Apigenin	Sustained release of drug at target site with improved AUC and delayed liver clearance	Increased accumulation of nanoparticles in HepG2, Huh-7, and liver tissue in rats	[196]
Camptothecin	Provided higher uptake rate and accumulation in HepG2 cells	CD147 monoclonal antibody	[197]
Curcumin	Stability and aqueous solubility of curcumin were increased by several fold	Targeting HepG2 cells was achieved due to presence of galactose groups	[198]
Farnesol and cisplatin	Exhibited faster drug mobility, sustained particle release, site-specific action, and higher percentage of apoptotic death compared with single drug treatment	ROS generated DNA damage in HepG2 cells	[199]
Quercetin, ellagic acid, and gallic acid	Nanoformulation offered controlled release of bioactives with improved bioavailability	Induced apoptosis-mediated cell death in HepG2 cells	[200]
Umbelliferone β-D-galactopyranoside	Effectively mitigated diethyl nitrosamine-induced HCC as confirmed through both histopathological and biochemical assays.	High hepatic accumulation of drug in rat model	[201]
Ursolic acid	Inhibited the growth of H22 xenograft and prolonged the survival time of tumor-bearing mice	Specific targeting or cellular uptake study was not performed	[202]
**Metallic-based nanoparticles ** **[203]**			
Hesperetin	Suppression of tumor necrosis factor alpha, transcription factor NF-κB, glycoconjugates, and proliferating cell nuclear antigen	Though specific targeting was not performed, the nanoparticles arrested DNA replication at late G1- and early S-phase	[204]
Resveratrol	Suppressed of tumor growth, promoted apoptosis, and decreased the expression of vascular endothelial growth factor.	Accumulation of nanoparticles in liver tissue was reported, along with apoptosis of cancer cells through PI3/Akt pathway	[205]
Epigallo- catechin gallate (EGCG)	Nanocages irradiated by NIR significantly upregulated caspase-3 by nearly two-fold and downregulated B-cell lymphoma 2 and caused cell apoptosis	Induced cancer cell apoptosis through changes in mitochondrial activities	[206]

## Data Availability

Not applicable.

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
