# Peer review of "Critical Review in Designing Plant-Based Anticancer Nanoparticles against Hepatocellular Carcinoma"

_pharmaceutics, 2023, doi:10.3390/pharmaceutics15061611_

Round 1
Reviewer 1 Report
I appreciate to authors for their extensive review on "Critical review in designing plant-based anticancer nanoparticles against hepatocellular carcinoma". I am agreed with the view of the authors that HCC is now a major challenge for human health because the number of patients is gradually increased. For the management of HCC natural products may be a good alternative. Nanotechnology added potential for the treatment of HCC. Nano-based delivery of natural products/ plant-based active ingredient may be a good option for HCC treatment. The manuscript was nicely written and sufficient data were given to show the impact of nano-based delivery of natural compounds for the treatment of HCC. A few minor corrections are required:
1. In the caption of Figure 2 reference is given Böttger R. et al. It needs reference numbering.
2. In Table 2 name of the plant Bistorta amplexicaulis will be in italics.
Author Response
Response to reviewer 1 comment: (please also see the attachment)
Comment 1: I appreciate to authors for their extensive review on "Critical review in designing plant-based anticancer nanoparticles against hepatocellular carcinoma". I am agreed with the view of the authors that HCC is now a major challenge for human health because the number of patients is gradually increased. For the management of HCC natural products may be a good alternative. Nanotechnology added potential for the treatment of HCC. Nano-based delivery of natural products/ plant-based active ingredient may be a good option for HCC treatment. The manuscript was nicely written and sufficient data were given to show the impact of nano-based delivery of natural compounds for the treatment of HCC. A few minor corrections are required: 1. In the caption of Figure 2 reference is given Böttger R. et al. It needs reference numbering.
Response: Thank you very much for your feedback and the comments on the manuscript. We have inserted the reference no. 49 in figure legend of figure 2 after Böttger R. et al. (see in page 6, line 209 (grey highlighted)).
Comment 2: 2. In Table 2 name of the plant Bistorta amplexicaulis will be in italics.
Response: Thank for the comment, the scientific name of Bistorta amplexicaulis in table 2 has now been in italic (see page 18, reference 186 (grey highlight)).

Reviewer 2 Report
This review was written and organized very well. I only have some corrections/suggestions/comments:
1) L238: "nanoparticle-protein corona" give a definition.
2) L245: "HCC tumor microenvironment" one of most important components of the tumor microenvironment is the cancer cell derived exosomes. However, authors did not refer to them (PMID: 30224923)
3) L246: "metabolic activities" add "(Figure 3A) at the end of this sentence.
4) L248: "Similarly, protons" Add some more details on these protons.
5) L256-258:"Non-invasive fusion technologies such as photodynamic, photothermal and hyperthermia treatments allow nanoparticles to generate cytotoxic conditions for tumor ablation." Please rephrase using more meaningful words.
6) L287: "Bcl-2 (B-cell lyphoma 2)" change to " B-cell lyphoma 2 (Bcl-2. Apply to the whole manuscript.
7) L298: "Genes" these could be "Proteins"
8) In Fig.4: Why you put NF-κB in cell cycle arrest? It should move up to Inflammation.
9) Move "3.3. Cell cycle arrest" before "3.2. Oncogene inhibition and tumor-suppression gene expression" to match the sequential order in Fig.4.
10) L328: "210 to 470 nM" is this nM or uM? please double check.
11) Why authors did not discuss the anti-inflammatory role of plant bioactives as a putative mechanism for their anticancer effects?
12) Sorafenib kills cancer cells through the induction of oxidative stress. So there is a debate regarding the use of antioxidants to combat cancer. Authors should speculate about that under "3.4. Antioxidant effects"
13) "3.5. Interference in cell signaling pathways" where is this part in Figure 4? Please add it.
14) In Table 1 and throughout the whole manuscript, please italicized gene names.
15) In Table 1: "HepG2CR" is not a common type of cell line. Add the full name as follow HepG2 cisplatin-resistant (HepG2CR)
16) In Table 1: I suggest adding the name of plants from which these plant bioactives isolated. As much as you can. This will be helpful for readers.
17) Tabe 1: "Crocin" Ic50 was lost. Delete " in HepG2" at "23.15 μM in HepG2"
18) Table 1: "Targets matrix metalloproteinase-2 secretion" AND "Prevented migration, invasion, and proliferation through down-regulation of MMP-9" and others in the table. So add anti-migratory effect as a potential effect for bioactives.
19) Table 1: "Eriocitrin," delete "," and add IC50.
20) Table 1: "ER_36-EGFR-Her-2 feedback loop." check for typos.
21) Table 1: "Resveratrol" you added one IC50 for each cell line but you cited two different reports. Did the two articles mention the same IC50 if not, use average or range?
22) Table 1: "Thymoquinone" one cell line but you add 2 cell lines IC50??
23) Fig. 5 lacks the exosomes as a nanoparticle-based delivery system. There are plenty of publications in that regard. So this novel approach should be mentioned.
24) Add the size of NPs in Table 2, and arrange the bioactives alphabetically in each carrier as you previously did in Table 1.
25) Correct Table 2 format and display, I found a very large space (empty second column) that could be suitable for NPs size.
26) Table 2: Explain uncommon abbreviation "EGCG"
27) In "5. Some current challenges and future perspectives" authors should speculate about cancer could not be treated with these bioactives alone and they could be given together with standard chemotherapy. And give more examples of this combinatory therapy. We have to not mislead the readers.
28) Conclusion is very long and has several repeats from Introduction.
Author Response
Response to reviewer 2 comment:
This review was written and organized very well. I only have some corrections/suggestions/comments:
Comment 1 L238: "nanoparticle-protein corona" give a definition.
Response: Thank you very much for your feedback and the comments on the manuscript. The definition of nanoparticle – protein corona has been inserted as suggested (see in page 6 line 240-241 (magenta highlighted)).
Comment 2 L245: "HCC tumor microenvironment" one of most important components of the tumor microenvironment is the cancer cell derived exosomes. However, authors did not refer to them (PMID: 30224923)
Response: Thank you for the comment. The term “exosomes” has been included as part of HCC tumor microenvironment in section 2, tumor biology, on page 4 line 148. (magenta highlighted) as well as in section 5, Current challenges and future perspectives on page 20 line 665-667. (magenta highlighted)
Comment 3 L246: "metabolic activities" add "(Figure 3A) at the end of this sentence.
Response: Thank you for the comments. The sentence was modified from “The HCC tumor microenvironment also demonstrates peculiar characteristics such as hypoxia, extracellular acidosis, and high interstitial pressure due to a drastic increase in tumorous metabolic activities”. To “The HCC tumor microenvironment also demonstrates peculiar characteristics such as hypoxia, extracellular acidosis, and high interstitial pressure due to a drastic increase in tumorous metabolic activities (Figure 3).” Please see page 7 line 250 (magenta highlighted).
Comment 4 L248: "Similarly, protons" Add some more details on these protons.
Response: Comment from reviewer is appreciated. The word “acidic” or (H+) has been inserted in sentence “Similarly, protons released in the extracellular acidic environment amplifies the risk of metastasis and create resistance to various cancer chemotherapeutics [74]” (see page 7, line 252, magenta highlighted).
Comment 5 L256-258:"Non-invasive fusion technologies such as photodynamic, photothermal and hyperthermia treatments allow nanoparticles to generate cytotoxic conditions for tumor ablation." Please rephrase using more meaningful words.
Response: Thank for the comment. The phrase “Non-invasive fusion technologies” has been replaced by “physical modalities” for simplicity (page 7, line 260, magenta highlighted).
Comment 6 L287: "Bcl-2 (B-cell lyphoma 2)" change to " B-cell lymphoma 2 (Bcl-2. Apply to the whole manuscript.
Response: Thank for the comment. The term “Bcl-2 (B-cell lyphoma 2)” has been replaced by “B-cell lymphoma 2 ” throughout the manuscript (page 8, line 290 and 296; page 9 line310; and in table 2, magenta highlighted).
Comment 7 L298: "Genes" these could be "Proteins"
Response: Suggestion from reviewer is appreciated. Term “Genes” has been replaced with “Proteins” in text (page 9, line 301).
Comment 8 In Fig.4: Why you put NF-κB in cell cycle arrest? It should move up to Inflammation.
Response: Thank for the comment. Figure 4 is now corrected as suggested.Please see figure in the attached file.
Comment 9 Move "3.3. Cell cycle arrest" before "3.2. Oncogene inhibition and tumor-suppression gene expression" to match the sequential order in Fig.4.
Response: Suggestion from reviewer is appreciated. However, we intend to arrange the molecular mechanisms of plant bioactive in the descending order according to their importance as follows (Apoptosis induction (26%) > Oncogene inhibition and tumor-suppression gene expression (24%) > Cell cycle arrest (19%) > Antioxidant effects (13%) > Anti-angiogenesis and Interference in cell signaling pathways.
Comment 10 L328: "210 to 470 nM" is this nM or uM? please double check.
Response: Comment from honourable reviewer is appreciated. The values have been re-checked with reference (Lee, W.J.; Shim, J.-Y.; Zhu, B.T. Mechanisms for the Inhibition of DNA Methyltransferases by Tea Catechins and Bioflavonoids. Mol. Pharmacol. 2005, 68, 1018–1030.) and confirmed with the same values as previously mentioned in the text (see page 8).
Comment 11 Why authors did not discuss the anti-inflammatory role of plant bioactives as a putative mechanism for their anticancer effects?
Response: Thank for the comment. Sentence “NF-κB related signaling pathways influence cancer related inflammation, neoplasia, hyperplasia, metastasis and even chemoresistance” had already been mentioned in page 9 line 2---301. In addition, the sentence “Activation of NF-κB also cause release of inflammatory cytokines that modulate the tumor microenvironment.” has been inserted in page 9 line 303-304. However, since the manuscript is focusing mostly on the molecular pathways of HCC progenesis, inflammation is considered as a symptomatic phenomenon based on cytokine release (1,2). Therefore, detail discussion on this is beyond the scope of the manuscript.
References:
1. Vaz, C., Mer, A. S., Bhattacharya, A., & Ramaswamy, R. (2011). MicroRNAs modulate the dynamics of the NF-κB signaling pathway. PLoS One, 6(11), e27774.
2. Qiu, J., Zhang, T., Zhu, X., Yang, C., Wang, Y., Zhou, N. et al. (2019). Hyperoside induces breast cancer cells apoptosis via ROS-mediated NF-κB signaling pathway. International Journal of Molecular Sciences, 21(1), 131.
Comment 12 Sorafenib kills cancer cells through the induction of oxidative stress. So there is a debate regarding the use of antioxidants to combat cancer. Authors should speculate about that under "3.4. Antioxidant effects"
Response: The suggestion of the reviewer is interesting. Literatures suggested that ROS in cancer cells act as secondary messengers and are linked to tumor progression and migration (1). This can be prevented and minimized by plant bioactives. Combination treatment between plant bioactives and classical chemotherapeutics have been explored in various capacities, mainly to bring out a synergistic effect against cancer cells. In case of sorafenib, such effects depend on tyrosine kinase inhibition and pro-apoptotic activities of both sorafenib and the bioactives, rather than the induction of oxidative stress (2,3). Since section 3.4 is completely dedicated to plant bioactives, examples of sorafenib-plant bioactive (rutin, fisetin and silibinin) combinations have been addressed in section 5 (see page 20 line 618-621, magenta highlighted).
References:
- Kumar, B., Koul, S., Khandrika, L., Meacham, R. B., & Koul, H. K. (2008). Oxidative stress is inherent in prostate cancer cells and is required for aggressive phenotype. Cancer research, 68(6), 1777-1785.
- 2. Youssef, M. M., Tolba, M. F., Badawy, N. N., Liu, A. W., El-Ahwany, E., Khalifa, A. E., ... & Abdel-Naim, A. B. (2016). Novel combination of sorafenib and biochanin-A synergistically enhances the anti-proliferative and pro-apoptotic effects on hepatocellular carcinoma cells. Scientific reports, 6(1), 30717.
- Guney Eskiler, G., Deveci, A. O., Bilir, C., & Kaleli, S. (2019). Synergistic effects of nobiletin and sorafenib combination on metastatic prostate cancer cells. Nutrition and cancer, 71(8), 1299-1312.
Comment 13 "3.5. Interference in cell signaling pathways" where is this part in Figure 4? Please add it.
Response: Suggestion from the reviewer is appreciated. We have replaced terms “Change in signaling pathway” with “Interference in signaling pathway” to match the text (see page 8, magenta highlighted).
Comment 14 In Table 1 and throughout the whole manuscript, please italicized gene names.
Response: Suggestion gene names have been checked and made in italic throughout manuscript.
Comment 15 In Table 1: "HepG2CR" is not a common type of cell line. Add the full name as follow HepG2 cisplatin-resistant (HepG2CR)
Response: Suggestion from the reviewer is appreciated. We have replaced terms “HepG2CR” with “HepG2 cisplatin-resistant (HepG2CR)” to match the text (table 1 in page 11, magenta highlighted).
Comment 16 In Table 1: I suggest adding the name of plants from which these plant bioactives isolated. As much as you can. This will be helpful for readers.
Response: Thank for the comment. All biological sources (scientific names) for the bioactives are listed in column 1 of table 1 (see page11, table 1).
Comment 17 Tabe 1: "Crocin" Ic50 was lost. Delete " in HepG2" at "23.15 μM in HepG2"
Response: The IC50 of crocin against both HepG2 and HCCLM3 cell lines has not been reported in literatures and it is therefore unavailable. Similar cases of IC50 unavailability have been indicated as “-” throughout table 1. The phrase “in HepG2” is deleted from table 1 as suggested (see curcumin in table 1, page 12).
Comment 18 Table 1: "Targets matrix metalloproteinase-2 secretion" AND "Prevented migration, invasion, and proliferation through down-regulation of MMP-9" and others in the table. So add anti-migratory effect as a potential effect for bioactives.
Response: We thank the reviewer for the suggestion. Prevention of migration, invasion and proliferation of cancer cells has already been included as part of anti-angiogenesis (please see page 10, line 377 magenta highlighted).
Comment 19 Table 1: "Eriocitrin," delete "," and add IC50.
Response: Information regards the IC50 value of Eriocitin has never been according to literature review, and thus we could not provide this information (see page 12 table 1).
Comment 20 Table 1: "ER_36-EGFR-Her-2 feedback loop." check for typos.
Response: Typing error has been rectified. Changed from ER_36-EGFR-Her-2 to ER-36-EGFR-Her-2 (see Epigallocatechin-3-Gallate in table1, page 12).
Comment 21 Table 1: "Resveratrol" you added one IC50 for each cell line but you cited two different reports. Did the two articles mention the same IC50 if not, use average or range?
Response: The query from honorable reviewer is appreciated. We would wish to clarify that the first reference/report (reference no. 147 in the manuscript) is focused on the mechanism of resveratrol, and did not report any IC50 values. Therefore, the mentioned IC50 values have been collected from the second reference (reference no. 148; see page 13)
Comment 22 Table 1: "Thymoquinone" one cell line but you add 2 cell lines IC50??
Response: Thank for the comment. Information of 2 cell lines have been mentioned according to reviewer suggestion (see Thymoquinone in table 1, page 13).
Comment 23 Fig. 5 lacks the exosomes as a nanoparticle-based delivery system. There are plenty of publications in that regard. So this novel approach should be mentioned.
Response: The comments from the reviewer is appreciated. However, figure 5 (see page 15) and section 4 of the manuscript were decided to recruit only nanoparticles which have been (i) approved by the FDA to be used in clinical setting, and (ii) used as carriers for plant bioactives in preclinical study. Although exosomes are very interesting as upcoming drug delivery system, they have not been approved by FDA as drug carriers and still under clinical trials (1). Furthermore, reports on encapsulation of plant bioactives inside exosomes are scare. According to suggestions from honourable reviewer, we therefore mention this approach in section 5 with references (see under Current challenges and future perspectives, page 21, line 664 – 666, magenta highlighted).
References:
1. Rezaie, J., Feghhi, M., Etemadi, T. (2022). A review on exosomes application in clinical trials: Perspective, questions, and challenges. Cell Communication and Signaling, 20(1), 1-13.
Comment 24 Add the size of NPs in Table 2, and arrange the bioactives alphabetically in each carrier as you previously did in Table 1.
Response: Thank for the comment. All bioactives have been alphabetically arranged as suggested by honourable reviewer. Although size plays an important role in modulating the mechanism of the nanomedicines, so does other physicochemical parameters including surface charge density, surface chemistry and presence of targeting ligands. There is subtle difference in size of nanoparticles in each category in table 2 (see also in section 4 as mentioned below). Here, we want to highlight the design or formulation of nanocarriers, and the interaction between the plant bioactives and carrier system; information provided in table 2 has been used to assist these 2 aspects. In fact, consideration only particle size may lead to a biased purview of the therapeutic effects of nanofomulation presented in the table 2. Furthermore, table 2 compiles the plant bioactives that have been encapsulated in the carriers, to provide the readers an idea about what more could be explored in relation to table 1. According to suggestion from reviewer, a generalized size range of the described nanoparticles have been added in section 4 (page 15, line 485;page 16, line 514 and 543; page 17, line 571-572; magenta highlighted).
Comment 25 Correct Table 2 format and display, I found a very large space (empty second column) that could be suitable for NPs size.
Response: Thank you for the comment. Format of table 2 has been corrected. Please see table 2 on page 17-19.
Comment 26 Table 2: Explain uncommon abbreviation "EGCG"
Response: The full term of “Epigallocatechin gallate (EGCG)” has been added in table 2 (see page19).
Comment 27 In "5. Some current challenges and future perspectives" authors should speculate about cancer could not be treated with these bioactives alone and they could be given together with standard chemotherapy. And give more examples of this combinatory therapy. We have to not mislead the readers.
Response: Complied. An example of plant derived bioactive that has been used as standard chemotherapeutic is paclitaxel from Taxus brevifolia. Therefore, we can not speculate that cancer could not be treated with bioactive alone. However, we have added more examples of combination chemotherapy in page 20, lines 604-606, and lines 618-620 (magenta highlighted).
Comment 28 Conclusion is very long and has several repeats from Introduction.
Response: Thank you for the comment. The conclusion part has been shortened and the replication has been removed. Please see conclusion part page 21-22.

Reviewer 3 Report
The present review described the mechanisms of action of promising plant bioactive compounds against hepatocellular carcinoma (HCC) and highlighted the recent advancements in plant-based nanoparticle formulations for HCC treatment. Nanocarriers that have been employed to encapsulate both pure bioactives and crude extracts for application in various HCC models have been collected and compared. Finally, current limitations in nanocarrier design, challenges related to HCC microenvironment and future opportunities are also discussed for the clinical translation of plant-based nanomedicines from bench to bedside. The manuscript should be revised taking in consideration the following comments:
LLiposomal-based nanoparticles means Lipid nanoparticles? Liposomes can be considered as an early version of liposomal-based nanoparticles. Chapter 4.1 should be modified.
cBiocompatibility of nano-biomaterials for clinical trials should be also discussed.
Author Response
Response to reviewer 3 comment: (Please see the attached file)
The present review described the mechanisms of action of promising plant bioactive compounds against hepatocellular carcinoma (HCC) and highlighted the recent advancements in plant-based nanoparticle formulations for HCC treatment. Nanocarriers that have been employed to encapsulate both pure bioactives and crude extracts for application in various HCC models have been collected and compared. Finally, current limitations in nanocarrier design, challenges related to HCC microenvironment and future opportunities are also discussed for the clinical translation of plant-based nanomedicines from bench to bedside. The manuscript should be revised taking in consideration the following comments:
Comment 1: LLiposomal-based nanoparticles means Lipid nanoparticles? Liposomes can be considered as an early version of liposomal-based nanoparticles. Chapter 4.1 should be modified.
Response: Section 4.1 describes liposomes-based drug delivery systems and their associated derivatives including niosomes, transfersome and nano-liposomes. Although they do not necessarily come in nano-range, all of them have vesicular structures as opposed to lipid or solid-lipid nanoparticles. We thank honourable reviewer for pointing out the ambiguity and re-phrase “Liposomal-based nanoparticles” to “Liposomes and their derivatives”. We hope that this would reflect all the lipid-based vesicles we addressed in this review. (see page 15 line 481 and figure 5 in the attached file, green highlighted)
Comment 2: cBiocompatibility of nano-biomaterials for clinical trials should be also discussed.
Response: Biocompatibility is one of major challenges to clinical translation of nano-biomaterials and the issue of nanotoxicity has already been mentioned in page 21, line 655-659. According to suggestions from the reviewer, the issue is further elaborated with relation to clinical trials (see page 21, green highlighted).

Round 2
Reviewer 2 Report
Authors responded to all my comments.
Reviewer 3 Report
The manuscript can be published in the present form
No comments